# A Kinetic Model for the Modification of Al_2_O_3_ Inclusions during Calcium Treatment in High-Carbon Hard Wire Steel

**DOI:** 10.3390/ma14051305

**Published:** 2021-03-09

**Authors:** Zuobing Xi, Changrong Li, Linzhu Wang

**Affiliations:** School of Material and Metallurgy, Guizhou University, Guiyang 550025, China; bingzxi@163.com

**Keywords:** high-carbon hard wire steels, inclusions, calcium treatment, unreacted shrinking core model

## Abstract

Laboratory-scale experiments for the modification of Al_2_O_3_ inclusions by calcium treatment in high-carbon hard wire steel were performed and the compositions and morphological evolution of inclusions were studied. The kinetics of the modification of Al_2_O_3_ inclusions by calcium treatment were studied in high-carbon hard wire steel based on the unreacted shrinking core model, considering the transfer of Ca and Al through the boundary layer and within the product layer, coupled with thermodynamic equilibrium at the interfaces. The diffusion of Al in the inclusion layer was the limiting link in the inclusion modification process. The Ca concentration in molten steel had the greatest influence on the inclusion modification time. The modification time for inclusions tended to be longer in the transformation of higher CaO-containing calcium aluminate. The modification of Al_2_O_3_ into CA_6_ was fastest, while the most time was needed to modify CA into C_12_A_7_. It took about six times time longer at the later stage of inclusion modification than at the early stage. The complete modification time for inclusions increased with the square of their radii. The changes of CaO contents with melting time were estimated based on a kinetic model and was consistent with experimental results.

## 1. Introduction

High-carbon hard wire steels are mainly applied in massive engineering projects such as bridges, cables, airports, power stations, and dams [1,2,3]. High-carbon hard wire products are drawn into filaments with a diameter of about 5 mm [4,5]. The hard inclusions with large sizes affect the steel yield and performance. Al_2_O_3_ inclusions are brittle with high melting temperatures, which are the major factors believed to impact the performance of high-carbon hard wire steels [6]. Therefore, Ca is usually used to modify Al_2_O_3_ inclusions to improve the performance of high-carbon hard wire steels.

Calcium treatment is one of the most commonly used and effective methods for modifying non-metallic inclusions into liquid ones [7,8,9,10]. The solid alumina can be converted into calcium aluminate inclusions partially or completely during calcium treatment [11,12], reducing the blockage of the immersion nozzle during continuous casting [10,13]. Since the 1990s, many studies have been conducted to understand the modification mechanism for alumina inclusions using calcium treatment [9,14,15,16]. Research studies on the modification kinetics for Al_2_O_3_ inclusions have been conducted in order to evaluate the accurate addition amounts for calcium and to understand the modification evolution process. Lu et al. [17] first established the kinetic model for oxide and sulfide inclusions in the calcium treatment process. They assumed that the internal diffusion rate of the inclusions is extremely fast and the interface reaction is rapid, and they developed an inclusion evolution model. Higuchi et al. [18] revised the kinetic model for modification of Al_2_O_3_ inclusions by using a first-order reaction equation. They studied the gasification rate of calcium from the melt and the reaction rate between the melt and inclusions. In this model, they assumed that the size and number of inclusions remained constant. Visseret al. [19] divided the ladle into two reaction zones: one is a high-calcium and low-oxygen zone, while the other is a low-calcium and low-oxygen zone. The kinetics of calcium treatment in the ladle stage was simulated, in which the results were in agreement with the experimental results. Ito et al. [20] studied the factors affecting the kinetics of calcium treatment using laboratory experiments. The modification of inclusions by calcium treatment was improved by shortening the aluminum deoxidation time, increasing the gas stirring, and increasing the reaction time after calcium treatment. They compared the calculated results based on the unreacted nucleus model and experimental results, and then proposed that the limiting link for the inclusion modification reaction is the mass transfer process in the product layer. Han et al. [21] believed that the decisive step is the chemical reaction rate between alumina and liquid calcium aluminate. Park et al. [22] believed that Al_2_O_3_ inclusions can be treated as unreacted nuclei at the beginning of modification and that the limiting link of the modification process is the diffusion of Al in the inclusion layer. However, they only discussed the rationality of the model, while a complete dynamic model has not yet been established. Zhang et al. [12] proposed a kinetic model of inclusion modification, considering the reduction of calcium in slag, the calcium dissolution rate in steel, mass transfer in the boundary layer, and solute diffusion in the product layer. The transformation model for alumina to magnesia–alumina spinel inclusions was established by Galindo et al. [23], and it was found that the transformation of inclusions was affected by the reaction at the slag–metal interface. Tabatabaei et al. [24] developed a kinetic model of inclusion transformation, which was applied to study the slag–steel reaction in a ladle furnace and to predict the composition changes for steel and slag and the evolution of inclusions during Ca treatment. Turkdogan et al. [25] found that the size of inclusions affected the modification rate, whereby large-sized inclusions were more difficult to modify than small-sized inclusions. Ye et al. [26] and Zheng et al. [14] proposed that with the increase in the calcium content in molten steel, the modification route was:(1)$${\mathrm{Al}}_{2}O_{3}\to \;CaO\cdot 6Al_{2}O_{3}(CA_{6})\;\to \;CaO\cdot 2Al_{2}O_{3}(CA_{2})\;\to \;CaO\cdot Al_{2}O_{3}(CA)\;\to \;12CaO\cdot 7Al_{2}O_{3}(C_{12}A_{7})\;\to \;3CaO\cdot Al_{2}O_{3}(C_{3}A)$$

Numerical simulation has many advantages, such as operating at low temperatures, having good reproducibility, being low cost, and detailed experimental data being available. Therefore, more metallurgical workers are using numerical simulation methods to study the behavior of inclusions in steel, and then to obtain the variation law for each parameter in the process and the quantitative relationships between each parameter [27].

On the basis of the multilayer unreacted core model for alumina inclusions, a step-by-step reaction kinetic model for the modification of Al_2_O_3_ inclusions by calcium treatment in high-carbon hard wire steel was established. The effects of Al, Ca, and O contents on the modification of high-carbon hard wire steel was studied during calcium treatment. The conversion ratio, radii of inclusions, calcium oxide contents in inclusions, and modification times were predicted. This work is helpful for understanding the inclusion modification process and for improving calcium treatment technology.

## 2. Experiment

### 2.1. Experimental Procedure

The tested steels were produced based on the chemical compositions of SWRH62A steel. Two heat experiments with different amount of deoxidants (A and B) were carried out in a tubular resistance furnace, as shown in Figure 1. The corundum crucible containing about 400 g pure iron, ferrosilicon alloy, and electrolytic manganese was placed in the furnace. Table 1 lists the compositions of raw materials used in the study. The temperature was increased to 1600 °C using electric heating. The added alloys and sampling procedures used in the experiments are shown in Figure 2. The aluminum alloy and Si–Ca alloy were added to the liquid steel for deoxidation at certain times. Calcium treatment was carried out after Al deoxidization. Four samples were sucked out using a quartz tube (ϕ 5 mm) and then quenched by insertion into a sodium chloride solution at different times after calcium addition (60, 180, 600, and 720 s). During the steel smelting process, the argon gas flow rate was maintained at 5 min/L.

### 2.2. Composition Analysis for Steels and Inclusions

The contents of calcium and aluminum in experimental steels were assessed using inductively coupled plasma–mass spectrometry (ICP–MS, Su Zhou, China) The contents of C, Si, Mn, and S in experimental steels were assessed using a direct-reading spectrometer (Q4-TASMAN, Brooke, Germany). The O contents in experimental steels were assessed using an inorganic oxygen–hydrogen tester.

The metallographic samples were ground using abrasive papers and then polished. The two-dimensional morphologies and compositions of inclusions in a cross section of each sample were analyzed using scanning election microscopy (ZeissΣIGMA+ X-Max20, Baden-Wurttemberg, Germany) and energy-dispersive spectrometry. About 30 inclusions were detected in each sample.

## 3. Results

### 3.1. Chemical Compositions of Steels

Table 2 lists the measured compositions of steels. A significant difference was found in the mass fractions of calcium between the two experiments due to the different amounts of Si-Ca alloy added. This showed that steel A had a high calcium content of 0.0025 mass%, while steel B had a low calcium content of 0.0017 mass%.

### 3.2. Compositions and Morphologies of Inclusions

The compositions of inclusions detected at 60, 180, 600, and 720 s after calcium addition are shown in Figure 3. The mass fractions of CaO in calcium aluminate inclusions were in the range of 7.75–29.23% at 60 s after calcium addition and the average mass fraction of CaO was 19.22% in steel A. This indicates that the main types of inclusions were CA_6_ and CA_2_ in steel A. The average mass fractions of CaO in calcium aluminate inclusions in steel A increased to 40.17%, 50.05%, and 60.43% after adding calcium for 180, 600, and 720 s, respectively. This indicates that the inclusions were modified into CA + C_12_A_7_, C_12_A_7_, and C_12_A_7_ + C_3_A gradually with the prolonging of the calcium treatment time. The mass fraction of CaO in calcium aluminate inclusions for steel B was less than that in steel A. The average mass fractions of CaO in calcium aluminate inclusions in steel B increased to 9.09%, 20.04%, 32.53%, and 40.12% after adding calcium for 60, 180, 600, and 720 s, respectively. This indicates that the inclusions were modified into CA_6_, CA_6_ + CA_2_, CA_2_ + CA, and CA + C_12_A_7_ gradually with the prolonging of the calcium treatment time.

The compositions, morphologies, and elemental mappings of typical inclusions detected in samples after calcium addition are shown in Figure 4 (steel A) and Figure 5 (steel B). Figure 4a shows that the typical inclusion CaO·2Al_2_O_3_ that formed after calcium treatment for 60 s was irregular in steel A. The CaO·Al_2_O_3_ that formed after calcium treatment for 180 s was similar to a hexagon, of which the edges tended to be smooth. Typical inclusions 12CaO·7Al_2_O_3_ and 3CaO·Al_2_O_3_ occurred in steel A at later stages of deoxidation and their morphologies tended to be spherical. Figure 4a shows the typical inclusions of CaO·6Al_2_O_3_, 3CaO·8Al_2_O_3_, CaO·2Al_2_O_3_, and CaO·Al_2_O_3_ formed in steel B after calcium treatment for 60, 180, 600, and 720 s, respectively. Their morphologies were irregular and their sharp corners tended to be disappeared.

## 4. Discussion

### 4.1. Dynamic Model

The mass transfer process and modification of Al_2_O_3_ inclusions in high-carbon hard wire steels can be described by the unreacted core model shown in Figure 6 based on experimental results. The modification process of Al_2_O_3_ inclusions can be described as shown in Figure 6 based on experimental results. The Al_2_O_3_ inclusion wrapped in the CA_6_ layer formed at the start of calcium treatment and Ca transferred into the unreacted core of Al_2_O_3_ through the CA_6_ layer. Therefore, the unreacted core of Al_2_O_3_ gradually decreased and the CA_6_ layer gradually thickened, then Al_2_O_3_ inclusions transformed into CA_6_ completely. With the diffusion of Ca and its increasing content, a complex inclusion with a core of CA_6_ wrapped in a CA_2_ layer formed and Ca transferred from the product layer (CA_6_ layer) to boundary layer (the interface between CA_2_ and CA_6_), resulting in the formation of a spherical inclusion with CA_2_. Similarly, the CA_2_ was transformed into CA, C_12_A_7_, and C_3_A step-by-step with the transfer of Ca and the chemical reaction between calcium aluminates. The assumptions in the current model are as follows:All inclusions in molten steel are spherical before and during the calcium treatment process;The temperature of molten steel is very high at 1600 °C, so the interfacial reaction is assumed to be in equilibrium;To simplify the discussion of the model, the concentrations of calcium, aluminum, and oxygen in molten steel are assumed to be constant;The diffusion of all substances in the liquid calcium aluminate layer is steady, which is in accordance with Fick’s first law.

Taking the transformation of CA inclusions into C_12_A_7_ as an example, the process is divided into the following three steps.
Ca in molten steel diffuses to the C_12_A_7_ layer–molten steel interface, for which the reaction formula is:
(2)$$Ca_{\left(s\right)}\to {\left[Ca\right]}_{\left(l\right)}$$[Ca] passes through the C_12_A_7_ liquid phase, diffuses to the CA layer, and reacts with it:(3)$$15[Ca]+33{\left(CaO\cdot Al_{2}O_{3}\right)}_{\left(s\right)}=4{\left(12CaO\cdot 7Al_{2}O_{3}\right)}_{\left(s\right)}+10[Al]$$At this time, the generated [Al] diffuses outward through the C_12_A_7_ liquid phase layer and enters into the molten steel.

Figure 7 is a schematic of the transformation process of CaO·Al_2_O_3_ into 12CaO·7Al_2_O_3_ inclusions. In the Figure 7, r_0_ represents the radius of C_12_A_7_ inclusion after complete modification, r represents the radius of the unreacted CA inclusion, and *l*_1_ and *l*_2_ represent the interface between CA inclusion and C_12_A_7_ inclusions, respectively. In the refining process, argon blowing and stirring are used. (Ca) and (Al) diffuse rapidly in molten steel and in the high-temperature reaction process. Therefore, the rate control link in the modification process for inclusions is solute diffusion in the calcium aluminate layer. At 1600 °C, the diffusion coefficients of Ca and Al in the calcium aluminate layer are [24,29] D_Ca_ ≈ 10^−8.6^ m^2^·s^−1^ and D_Al_ ≈ 10^−10.4^ m^2^·s^−1^, respectively. In this paper, the kinetics of the Al_2_O_3_ inclusion modification in high-carbon hard wire steel were analyzed in two cases. In the first case, the diffusion process of Al in the calcium aluminate layer was the limiting link. In the second case, the diffusion of Ca in the calcium aluminate layer was the limiting link in the process of inclusion modification.

When the diffusion of Al in the calcium aluminate layer is the limiting link in inclusion modification, the diffusion rate of Al in the C_12_A_7_ layer is expressed as:(4)$$v_{Al}=-\frac{dn_{Al}}{dt}=4\pi r^{2}D_{Al}\frac{dc_{{}^{{}_{Al}}}}{dr}$$
where *n_Al_* represents the amount of Al, *r* represents the radius of unreacted nuclear CA, *D_Al_* indicates the diffusion rate of Al in molten steel, *c_Al_* is the concentration of Al in C_12_A_7_, and *t* represents the modification time of inclusions:(5)$$dc_{{}^{{}_{Al}}}=-\frac{1}{4\pi D_{Al}}\frac{dn_{Al}}{dt}\frac{dr}{r^{2}}$$

Equation (5) is integrated as:(6)$$\int_{C_{A{l,l}_{1}}}^{C_{A{l,l}_{2}}}dc_{{}^{{}_{Al}}}=-\frac{1}{4\pi r^{2}D_{Al}}\frac{dn_{Al}}{dt}\int_{r}^{r_{0}}\frac{dr}{r^{2}}$$

From Equation (6), we get:(7)$$v_{Al}=-\frac{dn_{Al}}{dt}=4\pi D_{Al}\frac{r_{0}r}{r_{0}-r}(c_{Al,l_{1}}-c_{Al,l_{2}})$$

In the formula, $c_{{}^{{}_{Al,l_{1}}}}$ represents the Al concentration at the interface between two inclusions and $c_{{}^{{}_{Al,l_{2}}}}$ represents the Al concentration at the interface between inclusion C_12_A_7_ and molten steel.

According to Equation (7), the rate of Al consumption in the modification reaction is:(8)$$-\frac{dn_{Al}}{dt}=-\frac{2dn_{Al_{2}O_{3}}}{dt}=-\frac{x_{Al_{2}O_{3}}dn_{CA}}{dt}=-\frac{51}{79}\frac{4\pi r^{2}\rho_{CA}}{M_{CA}}\frac{dr}{dt}$$
where $\rho_{CA}$ represents the density of CA, $\rho_{CA}$ = 2.96 × 10^3^ kg/m^3^, and $M_{CA}$ represents the molar mass of CA, $M_{CA}$=158g/mol.

Combining Equation (7) with Equation (8), we can obtain:(9)$$\int_{0}^{t}\frac{79M_{CA}D_{Al}(c_{Al,l_{1}}-c_{Al,l_{2}})}{51\rho_{CA}}dt=\int_{r_{0}}^{r}\left(r-\frac{r^{2}}{r_{0}}\right)dr$$

The relationship between the unreacted nucleus radii of inclusions and the modification time (*t*) can be obtained by finishing the following:(10)$$t=\frac{17\rho_{CA}r_{0}^{2}}{158M_{CA}D_{Al}(c_{Al,l_{1}}-c_{Al,l_{2}})}\left[1-3{\left(\frac{r}{r_{0}}\right)}^{2}+2{\left(\frac{r}{r_{0}}\right)}^{3}\right]$$

To calculate the modification time of inclusions, the activity of solute elements in steel was used instead of its concentration. Equation (10) can be expressed as:(11)$$t=\frac{17\rho_{CA}r_{0}^{2}}{158M_{CA}D_{Al}(a_{Al,l_{1}}-a_{Al,l_{2}})}\left[1-3{\left(\frac{r}{r_{0}}\right)}^{2}+2{\left(\frac{r}{r_{0}}\right)}^{3}\right]$$

When CA inclusions are completely transformed into C_12_A_7_, when *r* = 0, the complete modification time (t*_f_*) of inclusions is:(12)$$t_{f}=\frac{17\rho_{CA}r_{0}^{2}}{158M_{CA}D_{Al}(a_{Al,l_{1}}-a_{Al,l_{2}})}$$
where $\rho_{CA}$, $M_{CA}$, and *D_Al_* are all constants and the modification time for CA inclusions depends on their radii and the activity difference of Al between the interface between two inclusions and the interface between molten steel and inclusions.

When the diffusion of Ca in the calcium aluminate layer is the limiting link in inclusion modification, the diffusion rate of Ca in the C_12_A_7_ layer is expressed as follows:(13)$$v_{Ca}=\frac{dn_{Ca}}{dt}=4\pi r^{2}D_{Ca}\frac{dc_{{}^{Ca}}}{dr}$$
where $n_{Ca}$ represents the amount of Ca, *r* represents the radius of unreacted nuclear CA, *D*_*C*a_ indicates the diffusion rate of Ca in molten steel, *c*_*C*a_ is the concentration of Ca in C_12_A_7_, and *t* represents the modification time of inclusions:(14)$$dc_{{}^{{}_{Ca}}}=\frac{1}{4\pi D_{Ca}}\frac{dn_{Ca}}{dt}\frac{dr}{r^{2}}$$

Equation (14) is integrated as:(15)$$\int_{C_{{Ca,l}_{1}}}^{C_{Ca{,l}_{2}}}dc_{Ca}=\frac{1}{4\pi r^{2}D_{Ca}}\frac{dn_{Ca}}{dt}\int_{r}^{r_{0}}\frac{dr}{r^{2}}$$

From Equation (15), we get:(16)$$v_{Ca}=\frac{dn_{Ca}}{dt}=4\pi D_{Ca}\frac{r_{0}r}{r_{0}-r}(c_{Ca,l_{1}}-c_{Ca,l_{2}})$$

In the formula, $c_{{}^{{}_{Ca,l_{1}}}}$ represents the Ca concentration at the interface between two inclusions and $c_{{}^{{}_{Ca,l_{2}}}}$ represents the Ca concentration at the interface between inclusion C_12_A_7_ and molten steel.

The rate of Ca generated by the modification reaction is:(17)$$\frac{dn_{Ca}}{dt}=\frac{2dn_{CaO}}{dt}=\frac{x_{CaO}dn_{C_{12}A}{}_{{}_{7}}}{dt}=\frac{16}{33}\frac{4\pi r^{2}\rho_{C_{12}A_{7}}}{M_{C_{12}A}{}_{{}_{7}}}\frac{dr}{dt}$$
where $\rho_{C_{12}A}{}_{{}_{7}}$ represents the density of C_12_A_7_, $\rho_{C_{12}A_{7}}$ = 2.83 × 10^3^ kg/m^3^, and $M_{C_{12}A_{7}}$ represents the molar mass of C_12_A_7_, $M_{C_{12}A}{}_{{}_{7}}$ = 1386g/mol.

Combining Equation (16) with Equation (17), we can obtain:(18)$$\int_{0}^{t}\frac{33M_{C_{12}A_{7}}D_{Ca}(c_{Ca,l_{1}}-c_{Ca,l_{2}})}{16\rho_{CA}}dt=\int_{r_{0}}^{r}\left(r-\frac{r^{2}}{r_{0}}\right)dr$$

The relationship between unreacted nucleus radii of inclusions and modification time (*t*) can be obtained by finishing:(19)$$t=\frac{8\rho_{C_{12}A}{}_{{}_{7}}r_{0}^{2}}{99M_{C_{12}A_{7}}D_{Ca}(c_{Ca,l_{1}}-c_{Ca,l_{2}})}\left[1-3{\left(\frac{r}{r_{0}}\right)}^{2}+2{\left(\frac{r}{r_{0}}\right)}^{3}\right]$$

To calculate the modification time of the inclusions, the activity of solute elements in steel was used instead of its concentration. Equation (18) can be expressed as:(20)$$t=\frac{8\rho_{C_{12}A}{}_{{}_{7}}r_{0}^{2}}{99M_{C_{12}A_{7}}D_{Ca}(a_{Ca,l_{1}}-a_{Ca,l_{2}})}\left[1-3{\left(\frac{r}{r_{0}}\right)}^{2}+2{\left(\frac{r}{r_{0}}\right)}^{3}\right]$$

When CA inclusions are completely transformed into C_12_A_7_, when *r* = 0, the complete modification time (*t_f_*) of inclusions is:(21)$$t_{f}=\frac{8\rho_{C_{12}A}{}_{{}_{7}}r_{0}^{2}}{99M_{C_{12}A_{7}}D_{Ca}(a_{Ca,l_{1}}-a_{Ca,l_{2}})}$$
where $\rho_{C_{12}A}{}_{{}_{7}}$, $M_{C_{12}A_{7}}$, and *D_C_*_a_ are all constants, and the modification time of CA inclusions depends on their radius and the activity difference of Ca between the interface between two inclusions and the interface between molten steel and inclusions.

### 4.2. Model and Parameter Determination

When the diffusion of Al in the calcium aluminate layer is the limiting link in inclusion modification, the concentration of [Al] in calcium aluminate inclusions is difficult to determine and can be replaced by a_Al_ as the following formula:

For interface *l*_1_:(22)$$15[Ca]+33{\left(CaO\cdot Al_{2}O_{3}\right)}_{\left(s\right)}=4{\left(12CaO\cdot 7Al_{2}O_{3}\right)}_{\left(s\right)}+10[Al]$$

The Gibbs free energy of this reaction is [25,29,30]:(23)$$\Delta G_{1}^{\theta }=-565,555-1609.66T,J\cdot mol^{-1}$$
(24)$$\Delta G_{1}{}^{\theta }=-RT\ln K^{\theta }=-RT\ln\frac{a_{Al,l_{1}}^{10}a_{{{}_{12}}_{CaO}{{}_{\cdot }}_{7Al_{2}O_{3}}}^{4}}{a_{Ca}^{15}a_{{}_{CaO\cdot Al_{2}O_{3}}}^{33}}$$
(25)$$a_{Al}{}_{,l_{1}}={\left[\frac{a_{Ca}^{15}a_{{}_{CaO\cdot Al_{2}O_{3}}}^{33}}{a_{{{}_{12}}_{CaO}{{}_{\cdot }}_{7Al_{2}O_{3}}}^{4}}\exp(\frac{\Delta G_{1}{}^{\theta }}{-RT})\right]}^{\frac{1}{10}}$$

In the formula, $a_{Al}{}_{,l_{1}}$ represents the activity of the interface layer Al between inclusion CA and C_12_A_7_, while $a_{{}_{CaO\cdot Al_{2}O_{3}}}^{}$ and $a_{{{}_{12}}_{CaO}{{}_{\cdot }}_{7Al_{2}O_{3}}}^{}$ are regarded as 1:(26)$$a_{i}=f_{i}\times [\%i]$$
(27)$$a_{Al}{}_{,l_{2}}=f_{Al}\times [\%\mathrm{Al}]$$
where $a_{Al}{}_{,l_{2}}$ represents the activity of Al at the interface between C_12_A_7_ inclusions and molten steel.

When the diffusion of Ca in the calcium aluminate layer is the limiting link in inclusion modification, the activity of the interface layer Ca between inclusion CA and C_12_A_7_ can be estimated as Equation (28).

For interface *l*_1_:(28)$$a_{Ca}{}_{,l_{1}}={\left[\frac{a_{Al}^{10}a_{{{}_{12}}_{CaO}{{}_{\cdot }}_{7Al_{2}O_{3}}}^{4}}{a_{{}_{CaO\cdot Al_{2}O_{3}}}^{33}\exp(\frac{\Delta G_{1}{}^{\theta }}{-RT})}\right]}^{\frac{1}{15}}$$

In the formula, $a_{Ca}{}_{,l_{1}}$ represents the activity of the interface layer Ca between inclusion CA and C_12_A_7_, while $a_{{}_{CaO\cdot Al_{2}O_{3}}}^{}$ and $a_{{{}_{12}}_{CaO}{{}_{\cdot }}_{7Al_{2}O_{3}}}^{}$ are regarded as 1.

For interface *l*_2_:(29)$$a_{Ca}{}_{,l_{2}}=f_{Ca}\times [\%\mathrm{Ca}]$$

The formula shows the activity of Ca at the interface between C_12_A_7_ inclusions and molten steel.

According to the interface shown in Equation (13), combined with the experimental steel composition in Table 2 and the interaction coefficient between elements in molten steel at 1873 K as shown in Table 3.

Equation (17) was used to calculate the activities of Ca and Al in each test steel (as shown in Table 4).

### 4.3. Determination of Restrictive Links

According to Equations (12) and (21), the diffusion of Al and Ca in the C_12_A_7_ layer, which treated as limiting link in the transformation of CA inclusions into C_12_A_7_ in steel A, was calculated as shown in Figure 8. It can be seen that the value of t_f_ is larger in the red line than that in the black line when the radius is same. This indicates that the diffusion of Al in the inclusion layer was the limiting link in the modification process of inclusions, which was consistent with the research results of Park et al. [24].

### 4.4. Effects of Solute Element Content in Molten Steel on Modification Time

The modification of inclusions depends on the activity of Al, which is affected by the content of each solute element in molten steel. The effects of Ca, Al, and O contents in molten steel on the time taken to modify CA into C_12_A_7_ inclusions with a radius of 1.5 μm was studied based on Equations (12), (25), and (27), as shown in Figure 9. The modification time increased linearly with increasing oxygen content and decreased with increasing calcium content in molten steel. When the content of O increased in the range of 0.0002–0.0045%, the modification time was prolonged from 0.0011 s to 118 s, increasing by approximately 118 s. When the content of Ca increased in the range of 0.0002–0.0045%, the modification time for complete transformation of Ca inclusions into C_12_A_7_ was reduced from 48616 s to 247 s, which was reduced by approximately 48369 s. The change of modification time with increasing Al content was very small. It can be seen that the Ca concentration in molten steel had the greatest influence on the modification time of inclusions. The steel with high Ca content was favorable for the modification of CA inclusions into C_12_A_7_ inclusions.

The relationship between the Ca content in molten steel and modification time of inclusions was calculated based on Equations (12) and (25), as shown in Figure 10. The complete modification time for inclusions was significantly shortened with the increase of the Ca content in molten steel and decrease of the inclusion radius. When the Ca content in molten steel was 0.0025% and the radius of inclusion was 1.5 μm, the modification times for Al_2_O_3_ into CA_6_, CA_6_ into CA_2_, CA_2_ into CA, CA into C_12_A_7_, and C_12_A_7_ into C_3_A were 4.5, 16, 116, 601, and 449 s, respectively. This indicates that the modification times for inclusions tend to be longer in the transformation of higher CaO-containing calcium aluminate. The modification of Al_2_O_3_ into CA_6_ was fastest, while the most time was required to modify CA into C_12_A_7_. When calcium aluminate inclusions changed from solid state to liquid state, this process was the most difficult to carry out and the required modification time was the longest.

### 4.5. Influence of Inclusion Conversion Rate in Molten Steel on Modification Time

In order to evaluate the modification rate, taking CA to C_12_A_7_ as an example, the inclusion conversion ratio is defined as:(30)$$\alpha =\frac{m_{CA(\mathrm{Initial})}-m_{CA(\mathrm{End})}}{m_{CA(\mathrm{Initial})}}\times 100\%$$

According to Equations (11) and (30), the relationship between the conversion ratio and modification time during inclusion deformation was calculated, as shown in Figure 11. The conversion ratio was affected by the melting time, inclusion type, and size. The conversion ratio increased quickly at the beginning of the modification but became slow as the reaction progressed. It can be seen that when the conversion ratio of CA to C_12_A_7_ with a 2 μm radius increased from 0% to 57.8%, this took 167 s, while the time needed to prolong the process was 902 s, with a conversion ratio increase from 57.8% to 100%; it took about six times longer time at the later stage of inclusion modification than at the early stage. Therefore, in the later stage of inclusion modification, the stirring speed should be increased to promote inclusion modification and reduce the modification time.

### 4.6. Relationship between Inclusion Radius and Modification Time

The relationship between the inclusion radius and modification time was studied based on Equation (12), as shown in Figure 12. The complete modification times for inclusions increased with the square of their radii. The complete modification time was prolonged by four times, while the radii of unmodified inclusions doubled. The complete modification time for inclusions with a 1 μm radius from CA to C_12_A_7_ was 267 s, and was 1069 s for inclusions with a 2 μm radius. During the whole modification process of solid Al_2_O_3_ inclusions to liquid calcium aluminate inclusions of the same size, the modification time for inclusions from CA to C_12_A_7_ was the longest.

### 4.7. Modeling Verification

In order to verify the calculation of the kinetic model for the modification of Al_2_O_3_ inclusions during calcium treatment, the calculated CaO content was compared with experimental results, considering the boundary conditions and unreacted core model parameters, as shown in Figure 13. The modification of inclusions with radii of 1, 2, and 3 μm was simulated based on observed results in experiments. It can be seen that the modification of Al_2_O_3_ inclusions in sample A is much faster than that in sample B, which is consistent with experimental results. The model calculation is closer to the experimental data, indicating that they are in good agreement. Additionally, it was found that the inclusions with 1 μm evolving from Al_2_O_3_ to CA_6_ took no longer than 1 s. It took about 1000 s for inclusions with a 3 μm radius to modify Al_2_O_3_ into liquid calcium aluminate in sample A and about 6000 s for that in sample B.

## 5. Conclusions

In the experiments with high-carbon hard wire steels treated with different contents of calcium at 1600 °C, the compositions and morphological evolution of inclusions were studied. The kinetic model for modification of Al_2_O_3_ inclusions during calcium treatment in high-carbon hard wire steels was established based on unreacted core theory. This model considers the transfer of Ca and Al through the boundary layer and within the product layer coupled with thermodynamic equilibrium at the interfaces. The calculated results based on the kinetic model were compared with experimental results. The main findings are summarized below:The diffusion of Al in the inclusion layer was the limiting link in the inclusion modification process. The modification time increased linearly with increasing oxygen content and decreased with increasing calcium content in molten steel. The change of modification time with increasing Al content was very small. The Ca concentration in molten steel had the greatest influence on the modification time of inclusions;The modification times for inclusions tended to be longer in the transformation of higher CaO-containing calcium aluminate. The modification of Al_2_O_3_ into CA_6_ was fastest, while the most time was needed to modify CA into C_12_A_7_;It took about six times time longer at the later stage of inclusion modification than at the early stage. The complete modification times for inclusions increased with the square of their radii. The complete modification times were prolonged by four times when the radii of unmodified inclusions doubled;The model calculation was in good agreement with experimental results. The inclusions with a 1 μm radius evolving from Al_2_O_3_ to CA_6_ took no longer than 1s. The modification of Al_2_O_3_ inclusions in sample A was much faster than in sample B. It took about 1000 s for inclusion with a 3 μm radius to modify Al_2_O_3_ into liquid calcium aluminate in sample A and about 6000 s for that in sample B.

## Figures and Tables

**Figure 1 materials-14-01305-f001:**
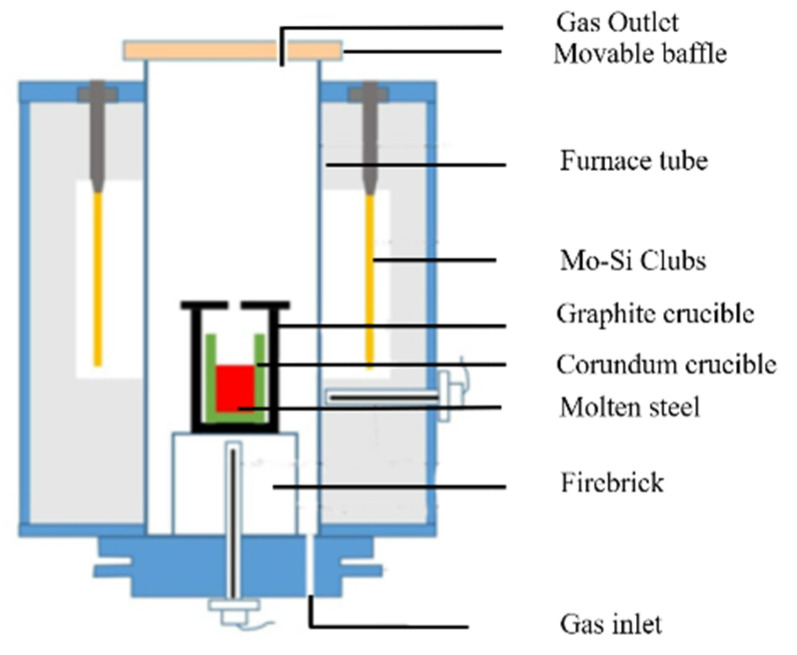
Schematic of tube furnace used in experiments. Reprinted with permission from [28]

**Figure 2 materials-14-01305-f002:**
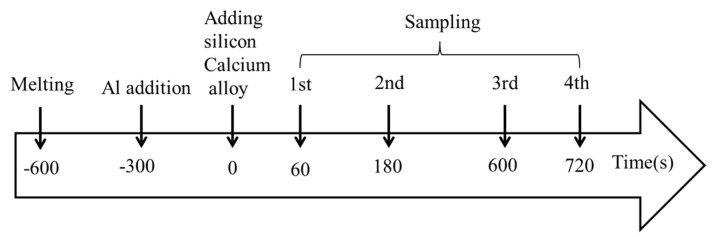
Addition of alloys and sampling procedure used in current experiments.

**Figure 3 materials-14-01305-f003:**
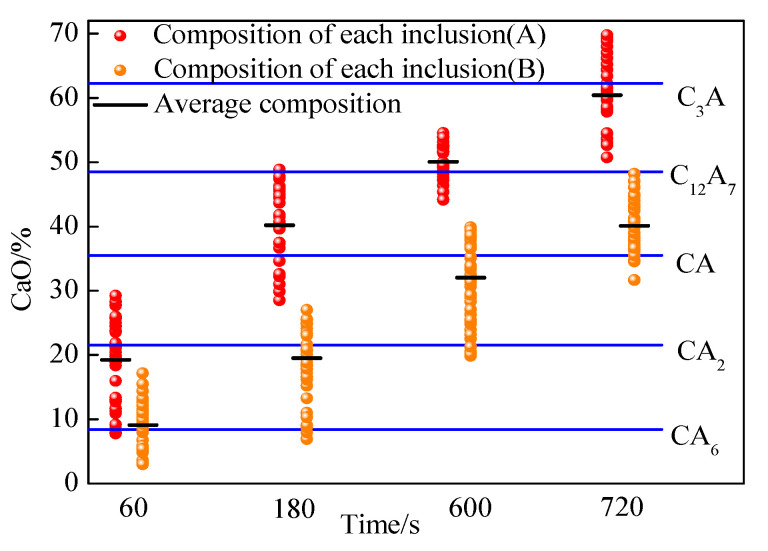
Compositions of inclusions detected in experiment steels (A and B) after calcium addition.

**Figure 4 materials-14-01305-f004:**
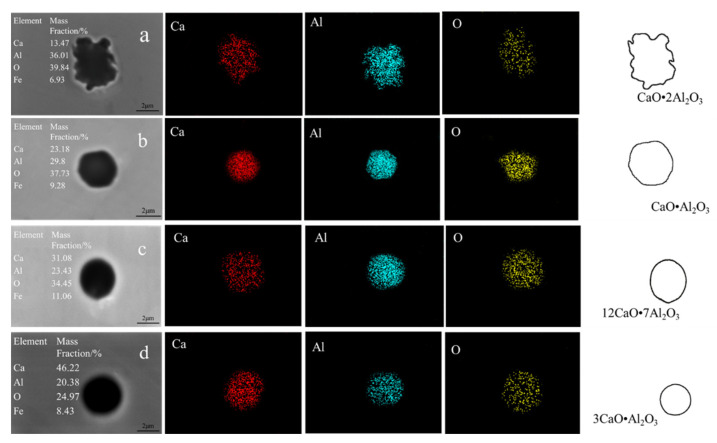
Elemental mappings of different types of inclusions detected after calcium additions in steel A: (**a**) 60 s; (**b**) 180 s; (**c**) 600 s; (**d**) 720 s.

**Figure 5 materials-14-01305-f005:**
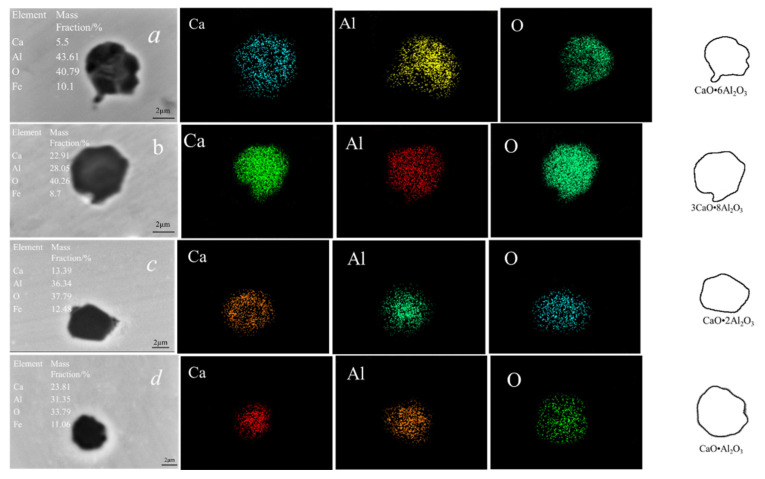
Elemental mappings of different types of inclusions detected after calcium additions in steel B: (**a**) 60 s; (**b**) 180 s; (**c**) 600 s; (**d**) 720 s.

**Figure 6 materials-14-01305-f006:**
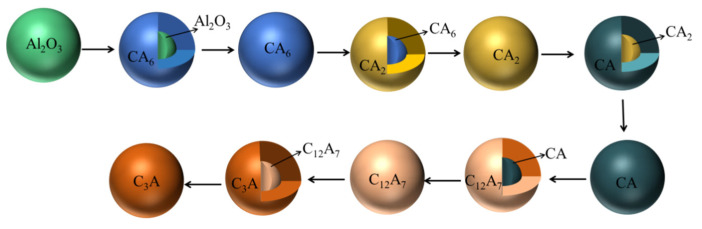
Kinetic model of alumina inclusions in calcium-modified steel.

**Figure 7 materials-14-01305-f007:**
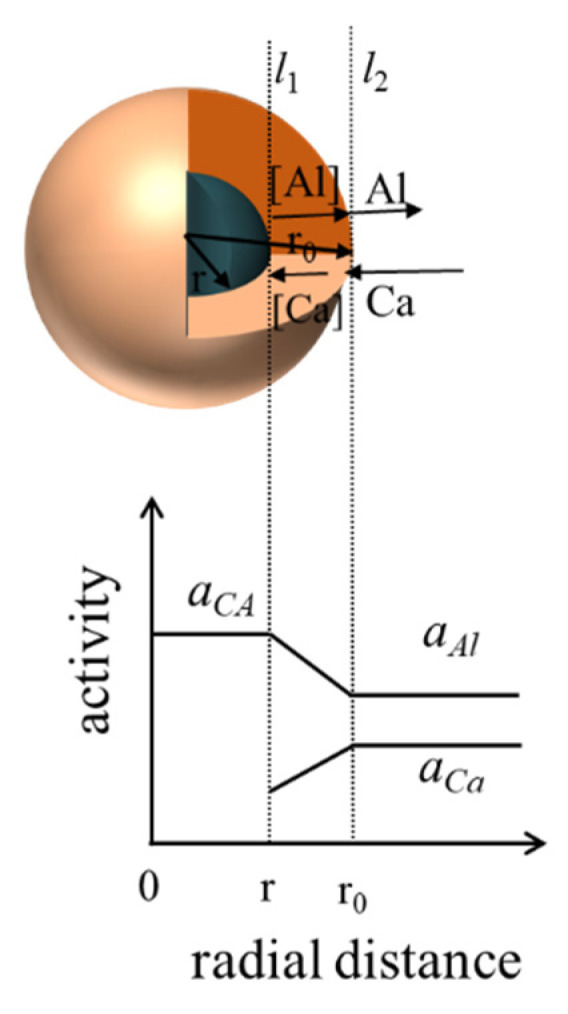
Schematic of the transformation of CA inclusions into C_12_A_7_ inclusions. (*l*_1_ represents the interface layer between CA and C_12_A_7_ inclusions, *l*_2_ represent that surface layer between the C_12_A_7_ inclusion and the molten steel boundary, *a*_CA_ indicates the activity of CA inclusions, *a_Al_* indicates the activity of Al, *a_Ca_* indicates the activity of Ca).

**Figure 8 materials-14-01305-f008:**
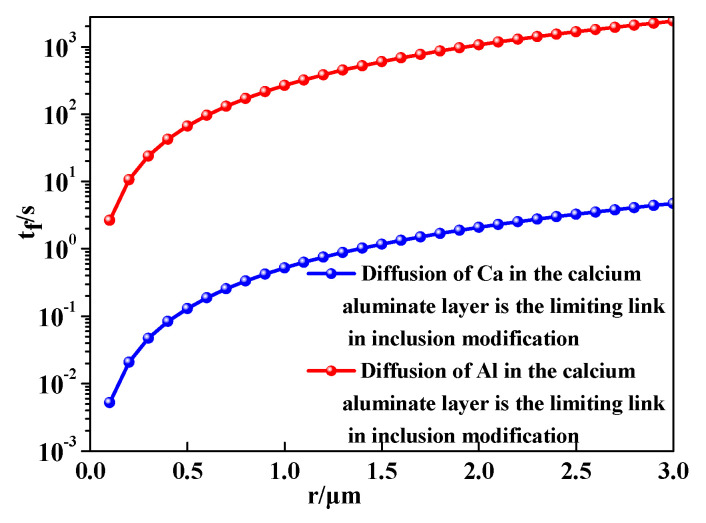
Diffusion of Ca and Al in the calcium aluminate layer is the limiting link of modification.

**Figure 9 materials-14-01305-f009:**
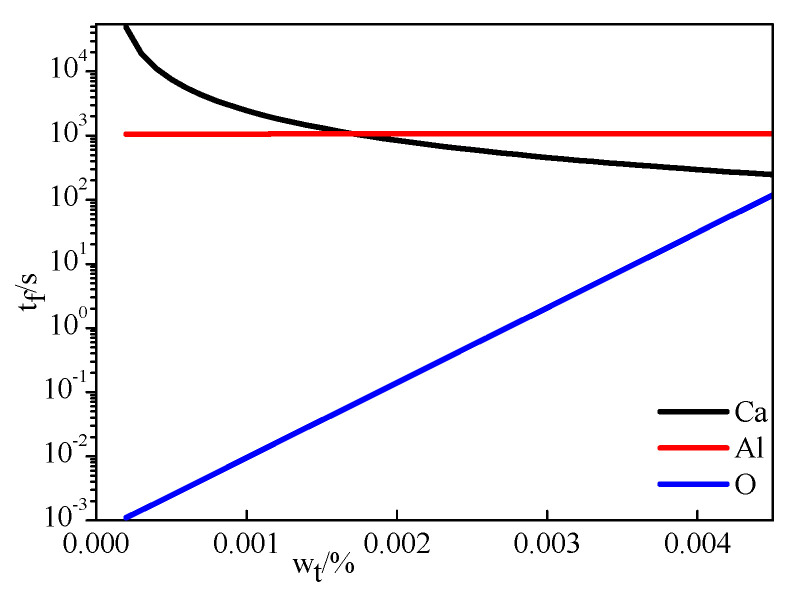
Effects of solute element content in molten steel on the time of complete modification of CA inclusions into C_12_A_7._

**Figure 10 materials-14-01305-f010:**
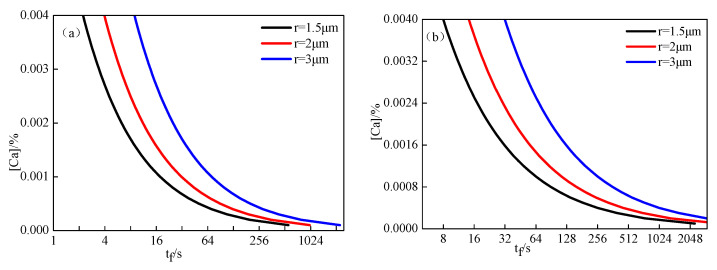
Relationship between Ca content in molten steel and complete modification times for inclusions with different compositions: (**a**) Al_2_O_3_ modified into CA_6_; (**b**) CA_6_ modified into CA_2_; (**c**) CA_2_ modified into CA; (**d**) CA modified into C_12_A_7_; (**e**) C_12_A_7_ modified into C_3_A.

**Figure 11 materials-14-01305-f011:**
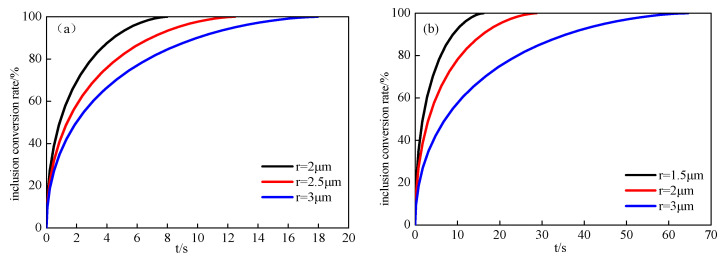
Modification process of inclusions with different particle sizes (**a**) Al_2_O_3_ is modified into CA6, (**b**) CA_6_ is modified into CA_2_, (**c**) CA_2_ is modified into CA, (**d**) CA is modified into C_12_A_7_, (**e**) C_12_A_7_ is modified into C_3_A).

**Figure 12 materials-14-01305-f012:**
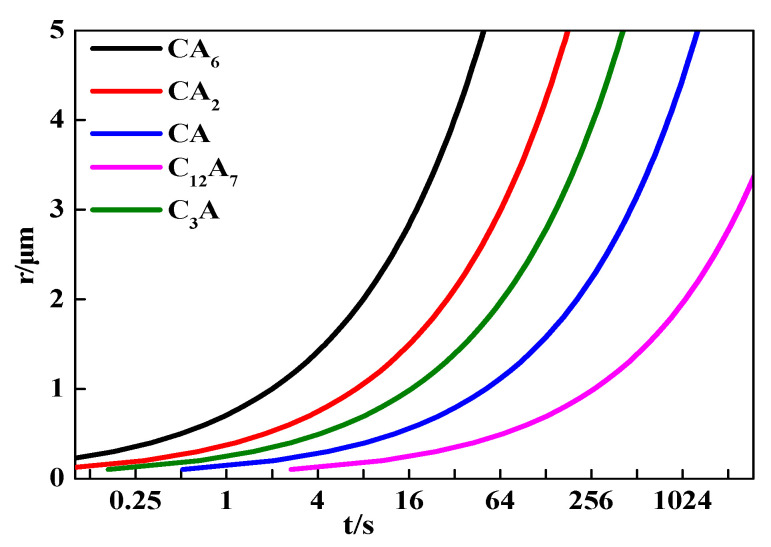
Complete modification time for Al_2_O_3_ inclusions of different sizes.

**Figure 13 materials-14-01305-f013:**
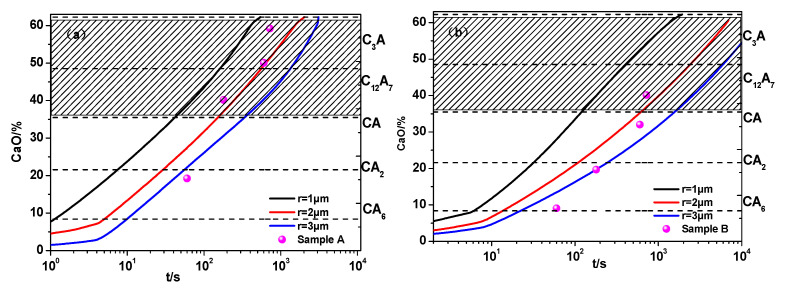
Variation of CaO contents in inclusions with time: (**a**) sample A; (**b**) sample B.

**Table 1 materials-14-01305-t001:** Compositions of raw materials (mass%).

Raw material	Fe	Si	Mn	S	C	Ca	Al	Others
Industrial pure iron	99.7	0.02	0.03	0.0002	0.0018	-	0.001	0.2445
Electrolytic manganese	-	-	99.999	-	-	-	-	0.001
Si–Fe alloy	21	78	0.4	0.02	0.1	-	-	0.48
Al alloy	0.7	0.8	0.15	-	-	-	96.94	1.41
Si–Ca alloy	-	57.13	20.44	-	0.83	19.56	2.02	0.02
QT400	95.8	0.17	0.5	0.01	3.45	-	-	0.07

**Table 2 materials-14-01305-t002:** Chemical compositions of steels used in different experiments (mass%).

Number	C	Si	Mn	S	O	Al	Ca
A	0.652	0.183	0.312	0.0021	0.0051	0.0042	0.0025
B	0.652	0.183	0.300	0.0017	0.0061	0.0038	0.0017

**Table 3 materials-14-01305-t003:** Interaction coefficients of Ca, O, and Al in molten steel at 1873 K (1600 °C) [13,31,32].

$e_{i}^{j}$	C	Si	Mn	S	Al	O	Ca
Ca	−0.34	−0.095	−0.007	−28	−0.072	−780	−0.002
O	−0.42	−0.066	−0.021	−0.13	−1.17	−0.17	−313
Al	0.091	0.056	−0.004	0.035	−0.043	−1.98	−0.047

**Table 4 materials-14-01305-t004:** Activity of Ca and Al in steel A and steel B.

Steel	*a_Ca_*	*a* _Al_
A	1.31 × 10^−7^	0.0048
B	1.52 × 10^−8^	0.0043

## Data Availability

Data is contained within the article.

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
