# Peer review of "A Kinetic Model for the Modification of Al2O3 Inclusions during Calcium Treatment in High-Carbon Hard Wire Steel"

_materials, 2021, doi:10.3390/ma14051305_

Round 1

Reviewer 1 Report

The article leaves the most positive impression upon a cursory reading and is written in the best traditions of a metallurgical experiment, interpreted by physicochemical calculations without such advance software as the diffusion module (DICTRA) is an add-on module to Thermo-Calc. It can be useful for both undergraduate and graduate students and practitioners. However, a detailed study of the article raises many questions, which are outlined below.

  1. at the beginning of line 124 there is a typo: you need to write C12A7 + C3A instead of C12A7 + C12A7
  2. lines 165-166: you should probably write a specific temperature, for example, 1600°C, for which the diffusion coefficients and interaction coefficients of Ca, O and Al were calculated.
  3. lines 167-168. It can be understood that the concentrations of aluminum and oxygen will be constant for a long time, since their content is much less than the solubility. But what about calcium? At 1550 and 1650°C the calcium concentrations in iron (contact with saturated Ca vapour!) are correspondingly 0.031 and 0.051 wt.%. See the paper [1] below. But a bubble of calcium vapor will linger near the inclusion only for a split second. In the absence of a bubble, the concentration of calcium around the inclusion will be negligible
  4. line 183: it is not clear where the values of l1 and l2 are shown in Fig. 6.
  5. table 4: column “Number” apparently have to be “Steel”.
  6. lines 283-286: the digits of fractions of a second, when it comes to hundreds or hundreds of thousands of seconds, do not matter, but they interfere with the perception of the text. The same remark is throughout the article.
  7. lines 287-288: “It can be seen that the Ca concentration in molten steel had the greatest influence on the modification time of inclusions” and Fig.8. Here again the question arises of how realistic this is in experiment, since the solubility of calcium in the melt is negligible and it is practically impossible to maintain it constant in the vicinity of the inclusion, which recrystallizes through the liquid melt. See above note #3.
  8. lines 297-299: The same question is as previous: how can the calcium concentration in the melt be kept at 0.0025% for 10 minutes in experiment? See above note #3.
  9. lines 301-304: it is not clear why diffusion proceeds faster in refractory solid hexa-aluminate than in low-melting calcium aluminates?
  10. lines 335-337: the same question is as previous. Solid phase diffusion is always slower.
  11. How to explain the relative position of the curves in Figure 11? It does not match the sequence in the series of modifications from CA6 to C3A (see Fig.5).
  12. Table 3: as follows from the paper [1]: Vagner parameter of the interaction of Ca and O in liquid iron should be eOCa = –1.41, not about hundreds, as yours.
  13. Literature has to be numbered.

Good agreement of experiment and calculation is a guarantee that the authors will answer the questions that have been arisen, so that the reader would be satisfied that the observed phenomena do not contradict fundamental knowledge.

Reference

[1] Mikhailov, G. G., and D. A. Zherebtsov. “On the Interaction of Calcium and Oxygen in Liquid Iron.” Materials Science Forum, vol. 843, Trans Tech Publications, Ltd., Feb. 2016, pp. 52–61. Crossref, doi:10.4028/www.scientific.net/msf.843.52

Author Response

  1. at the beginning of line 124 there is a typo: you need to write C12A7 + C3A instead of C12A7 + C12A7

Reply to reviewer: The errors pointed out by the reviewer have been corrected. Amendment (130)

  1. lines 165-166: you should probably write a specific temperature, for example, 1600°C, for which the diffusion coefficients and interaction coefficients of Ca, O and Al were calculated.

Reply to reviewer: According to your suggestion, the specific temperature have been added in the manuscript. Amendment (171)

  1. lines 167-168. It can be understood that the concentrations of aluminum and oxygen will be constant for a long time, since their content is much less than the solubility. But what about calcium? At 1550 and 1650°C the calcium concentrations in iron (contact with saturated Ca vapour!) are correspondingly 0.031 and 0.051 wt.%. See the paper [1] below. But a bubble of calcium vapor will linger near the inclusion only for a split second. In the absence of a bubble, the concentration of calcium around the inclusion will be negligible

Reply to reviewer: Compared with the whole process, the bubbles of calcium vapor exist for a short time. For the convenience of calculation, we think that the concentration of calcium in molten steel is constant.

  1. line 183: it is not clear where the values of l1 and l2 are shown in Fig. 6.

Reply to reviewer: According to your suggestion, the values of l1 and l2 have been added in the manuscript. As shown in Figure 7.

  1. table 4: column “Number” apparently have to be “Steel”.

Reply to reviewer: The errors pointed out by the reviewer have been corrected. As shown in Table 4.

  1. lines 283-286: the digits of fractions of a second, when it comes to hundreds or hundreds of thousands of seconds, do not matter, but they interfere with the perception of the text. The same remark is throughout the article.

Reply to reviewer: According to your suggestion, We have made changes in the manuscript.

  1. lines 287-288: “It can be seen that the Ca concentration in molten steel had the greatest influence on the modification time of inclusions” and Fig.8. Here again the question arises of how realistic this is in experiment, since the solubility of calcium in the melt is negligible and it is practically impossible to maintain it constant in the vicinity of the inclusion, which recrystallizes through the liquid melt. See above note #3.

Reply to reviewer: When O content in molten steel increases by 0.0043%, the complete denaturation time of inclusions increases by about 118s, and when Ca content in molten steel increases by 0.0043%, the complete denaturation time of inclusions decreases by about 48369s s. When Al content in molten steel changes, the complete denaturation time of inclusions has no significant change. According to the change of complete denaturation time of inclusions with the same amount of solute elements in molten steel, it is determined that the change of calcium concentration in molten steel has the greatest influence on the denaturation time of inclusions.

  1. lines 297-299: The same question is as previous: how can the calcium concentration in the melt be kept at 0.0025% for 10 minutes in experiment? See above note #3.

Reply to reviewer: The calcium concentration in the melt was not kept at 0.0025% for 10 minutes. After the experiment, the calcium content in the final test steel is determined to be 0.0025%, so this concentration is used for calculation.

  1. lines 301-304: it is not clear why diffusion proceeds faster in refractory solid hexa-aluminate than in low-melting calcium aluminates?

Reply to reviewer: Because the calcium content required for the transformation of Al2O3 into CA6 is less, and the calcium content required for the transformation of CA into C12A7 is more, it is found by calculation that the transformation of Al2O3 into CA6 is the fastest.

  1. lines 335-337: the same question is as previous. Solid phase diffusion is always slower.

Reply to reviewer: Although solid phase diffusion is slow, the content of Ca needed in the process of solid phase inclusion denaturation is less. It is found that the denaturation time needed in the process of solid phase inclusion denaturation is shorter by calculation and comparison.

  1. How to explain the relative position of the curves in Figure 11? It does not match the sequence in the series of modifications from CA6 to C3A (see Fig.5).

Reply to reviewer: In figure. 11, the black line indicates the denaturation time required when Al2O3 inclusions are denatured into CA6. The red line indicates the complete denaturation time required when CA6 inclusions are denatured into CA2 inclusions. The blue line indicates the complete denaturation time required when CA2 inclusions are denatured into CA inclusions. The pink line indicates the complete denaturation time required when CA inclusions are deformed into C12A7 inclusions. Green line indicates the complete denaturation time required for C12A7 inclusion to be denatured into C3A inclusion. The arrangement sequence of different color lines in figure. 11 indicates the required complete denaturation time, and does not represent the order of inclusion denaturation.

  1. Table 3: as follows from the paper [1]: Vagner parameter of the interaction of Ca and O in liquid iron should be eOCa = –1.41, not about hundreds, as yours.

Reply to reviewer: Table 3 lists the interaction coefficients of elements in molten steel. As follows from the paper[1].( [1] Zheng H Y, Guo S Q, Qiao M R, et al. Study on the modification of inclusions by Ca treatment in GCr18Mo bearing steel[J]. Advances in Manufacturing, 2019, 7(4):438-447. )

  1. Literature has to be numbered.

Reply to reviewer: Thanks for your suggestion. We added the literature number

Reviewer 2 Report

Please see attached review report. 

Author Response

1.Reply to reviewer: Multilayer unreacted core model is mostly used in the literature. In this paper, a stepwise reaction model is used to calculate the relationship between inclusion conversion rate and inclusion denaturation time. The modification time needed to modify solid alumina inclusions into liquid calcium aluminate inclusions is better described. Most of the research on the modification of inclusions in steel is in low carbon steel, but there is little research on the modification of alumina inclusions in high carbon hard wire steel.

2.Reply to reviewer: Thank you for your suggestion. We added pictures in the manuscript. As shown in figure 1.

3.Reply to reviewer: Thank you for your suggestion. We added the equipment model in the manuscript. The amendment is (96,108,113)

4.Reply to reviewer: Numerical simulation has many advantages, such as free from high temperature, good reproducibility, low cost and detailed experimental data. Therefore, more and more metallurgical workers use numerical simulation method to study the behavior of inclusions in steel, and then obtain the variation law of each parameter in the process and the quantitative relationship between each parameter.

5.Reply to reviewer: Thank you for your suggestion. We have checked and revised the manuscript, hoping to achieve better presentation effect.

Reviewer 3 Report

My comments are given in the following text. It is necessary to explain them in the text of the article and edit the text of the article.

row 28: High-carbon hard wire products are drawn into filaments with a diameter of 0.08–5.5 mm. .... row.32: Therefore, Ca is usually used to modify Al2O3 inclusions to improve the performance of high-carbon hard wire steels.

Q1: High-carbon steels intended for the production of wire with a diameter of 0.08–5.5 mm (eg. for the production of tire cords) are mostly produced without the addition of calcium and aluminum. The reason is simple - to prevent the formation of non-formable calcium aluminates, which are harmful during cold forming and also from the point of view of fatigue properties. Please explain the reason for the addition of calcium to these steels.

row. 52: The modification of inclusions by calcium treatment was improved by shortening the time of calcium treatment and aluminum deoxidation, increasing gas stirring, and increasing the reaction time after calcium treatment.

Q2: „shortening the time of calcium treatment“ and „increasing the reaction time after calcium treatment“ - Both factors are similar, why do they work in opposite ways?

row 86: Two heats of experiments with different amount of deoxidants (A and B) were carried out in a tubular resistance furnace.

Q3: I recommend supplementing the data and drawing of the experimental equipment, including the method of measuring the temperature, the weight of the steel in the crucible, the method of adding the Si-Ca alloy, the diameter of the aspirated samples, etc.

row 116: Compositions and morphologies of inclusions

Q4: I do not believe that the inclusions were only formed by oxides of aluminum and calcium. In my opinion, they also had to contain silicon oxides, because the silicon content in the steel was relatively high - 0.183%.

row 148: Discussion/Dynamic model 

Q5: The dynamic model seems to have been created with knowledge of mathematics and physical chemistry, and its conclusions are logical. But the conclusions only confirm the knowledge more than 20 years old about the mechanism of action of calcium on Al2O3 type inclusions.
In practice, however, the conditions for the transformation of Al2O3 inclusions are different than in a small crucible due to intensive stirring, time of treatment, rate of cooling during casting. The work also does not mention the possible influence of the ceramics of the crucible and suction tubes on the achieved results.

Author Response

1.Reply to reviewer: A major objective of calcium treatment in steelmaking is to improve product performance through inclusion modification. In high-carbon hard wire steel, it is not only beneficial to the prevention of nozzle blockage caused by alumina buildup during casting but also to modify inclusions to alleviate their deleterious effects on impact toughness and fracture toughness. In addition, lanthanum and cerium readily corrode the ladle refractories. When calcium treatment is efficiently performed, the alumina inclusions are converted to molten calcium aluminate which are globular in shape because of a surface tension effect. The calcium aluminate inclusions retained in liquid steel suppress the formation of MnS stringers during

solidification of steel[1][2].

([1] Abraham S , Bodnar R , Raines J . Inclusion engineering and metallurgy of calcium treatment[J]. Journal of Iron and Steel Research International, 2018, 1(2):1243-1257.

[2] Tanaka, Yasuhiro, Pahlevani, et al. Agglomeration Behavior of Non-Metallic Particles on the Surface of Ca-Treated High-Carbon Liquid Steel: An In Situ Investigation[J]. Metals, 2018.)

2.Reply to reviewer: The wrong statement in line 52 is that the time of calcium treatment is deleted by shortening the deoxidization time of aluminum, which has been corrected in the manuscript. Amendment (50)

3.Reply to reviewer: Thank you for your suggestion. We added the relevant data.As shown in figure 1,row 90 and row 96.

4.Reply to reviewer: During the energy spectrometer test, it was found that there were isolated silicon oxide inclusions in the test steel, but no compound calcium silicate inclusions were found.

5.Reply to reviewer: Thank you for your suggestion. In this experiment, the influence of the diffusion of molten elements, the content of molten elements in molten steel, inclusion radius and inclusion conversion rate on the modification time of inclusions was considered. 

Round 2

Reviewer 3 Report

Thank you for editing the text of the article. However, I still have doubts about the statement right at the beginning of the article:

row. 30: "Therefore, Ca is usually used to modify Al2O3 inclusions to improve the performance of high-carbon hard wire steels".

with connection of

row. 24: "High-carbon hard wire products are drawn into filaments with a diameter of 0.08–5.5 mm"

The authors should clearly explain whether in fact high carbon steels intended for the production of steel filaments with a diameter  below 0,2 mm (which can be cord steels for the production of tire reinforcements) can use a calcium additive. These filaments are produced by cold drawing and it is generally known that the presence of Al2O3-based inclusions but also xAl2O3.yCaO inclusions (which are non-deformable) causes serious problems in cold drawing these steels. 

These cord steels are usually produced without the addition of aluminum and calcium. If you have other experience with the technology of production of these steels, please state them incl. citation.

Author Response

High-carbon hard wire steel is often used in the production of high-end wire materials such as steel wire for bridge cables, automobile tire cord, steel strand and wire rope. In practice production, steel strand and wire rope will be deoxidized with Si-Al-Fe alloy[1],[2] . In this paper, the high carbon hard wire steel used to produce wire rope and steel strand is studied. In the production process of high carbon hard wire steel, wires will be drawn into filaments with a diameter of about 5 mm[3]. The existence of nonmetallic inclusions will affect the drawability of steel. The influence of brittle Al2O3 inclusions with high melting point is obvious. Therefore, Ca is usually used to modify Al2O3 inclusions to improve the performance of high-carbon hard wire steels[2],[4],[5]。

[1]Qin Junshan, Qu Tianpeng, Wang Deyong,et al. Mg treatment process for high carbon steel 82B [J].Steelmaking,2019,35(01):18-23+28.

[2]Gao Shengya, Jiang Min, Hou Zewang, et al. Effect of calciumtreatment on non-metallic inclusions in high carbon aluminum killed steel [J].Iron and Steel,2017,52(04):25-30.

[3]Dai Y , Li J , Shi C , et al. Influence of slag-changing operation during LF refining on the inclusions in 82B high carbon steel wire[J]. Metallurgical Research and Technology, 2019, 116(4):411.

[4]Parusov V V , Starov R V , Derevyanchenko I V , et al. A decade of quality steel production: Developing a production technology for steel used in metal-cord manufacture[J]. Steel in Translation, 2010, 40(1):82-87.

[5]Yasuhiro T , Farshid P , Karen P , et al. Engulfment Behavior of Inclusions in High-Carbon Steel: Theoretical and Experimental Investigation[J]. Metallurgical and Materials Transactions B, 2018, 49:2986-2997.